# Obesity-related indicators and tuberculosis: A Mendelian randomization study

**Nuannuan Cai[1], Weiyan Luo[1], Lili Ding[1], Lijin Chen[1], Yuanjiang Huang[2]***

1 Pulmonary and Critical Care Medicine, Hainan Provincial People's Hospital, Haikou, Hainan, China,
2 Infectious and Tropical Disease Dept (Tuberculosis), The Second Affiliated Hospital of Hainan Medical College, Haikou, Hainan, China

* huangyuanjiang@shhmu.net

**Data Availability Statement:** In this study, all data were derived from publicly available Open GWAS project database (https://gwas.mrcieu.ac.uk/), and all data can be obtained for free.

## Abstract

### Purpose

Obesity is a strong risk factor for many diseases, with controversy regarding the cause(s) of tuberculosis (TB) reflected by contradictory findings. Therefore, a larger sample population is required to determine the relationship between obesity and TB, which may further inform treatment.

### Methods

Obesity-related indicators and TB mutation data were obtained from a genome-wide association study database, while representative instrumental variables (IVs) were obtained by screening and merging. Causal relationships between exposure factors and outcomes were determined using two-sample Mendelian randomization (MR) analysis. Three tests were used to determine the representativeness and stability of the IVs, supported by sensitivity analysis.

### Results

Initially, 191 single nucleotide polymorphisms were designated as IVs by screening, followed by two-sample MR analysis, which revealed the causal relationship between waist circumference [odds ratio (OR): 2.13 (95% confidence interval (CI): 1.19–3.80); p = 0.011] and TB. Sensitivity analysis verified the credibility of the IVs, none of which were heterogeneous or horizontally pleiotropic.

### Conclusion

The present study determined the causal effect between waist circumference and TB by two-sample MR analysis and found both to be likely to be potential risk factors.

**Funding:** This study was supported by the impact of 4C continuity of care model on non-invasive ventilator use in patients with chronic obstructive pulmonary disease (Project number: 2101320712A2012). The funders had no role in study design, data collection and analysis, decision to publish, or preparation of the manuscript.

**Competing interests:** The authors have declared that no competing interests exist.

## Introduction

Tuberculosis (TB) is chronic infectious disease caused by the pathogenic bacterium *Mycobacterium tuberculosis (M. tuberculosis)* [1, 2], and it is one of the most common diseases of the lungs [3, 4]. Because the lungs are the organs most susceptible to *M. tuberculosis*, pulmonary involvement can affect > 80% of patients infected with *M. tuberculosis* [5]. Additionally, TB has an extremely high rate of infection and mortality, thereby posing a serious threat to human health [6, 7]. The pathogenesis of TB is slow and its involvement is widespread [8]. Typical symptoms include low-grade fever in the afternoon and night sweats, and atypical symptoms such as shortness of breath and weight loss [9]. Moreover, diagnosing TB in the clinic is difficult and often requires bacteriological methods [10]. Therefore, there is an urgent need to stress prevention and identify risk factors for TB to reduce the risk for transmission and death.

Obesity has long been a serious threat to human health worldwide, leading to multiple health problems [11]. The harm of obesity is mainly reflected in the human body's metabolic disorders [12], which leads to endocrine disorders and energy imbalance, and then produce the corresponding diseases including diabetes, hypertension, cardiovascular disease, and even increase the risk of tumors [13, 14]. In clinical practice, obesity-related indicators include body mass index (BMI), waist circumference, BMI-adjusted waist-to-hip ratio, hip circumference, arm circumference, blood lipid level, fat mass and low lean body mass [15, 16]. What's more, in recent years, there have been observational studies that suggest that being overweight also leads to an increased risk of developing latent TB [17]. This is quite different from previous studies, which have shown that being underweight or having a low BMI is a risk factor for active TB [18, 19]. The evidence on the effect of obesity on tuberculosis is not well documented, and the association between low BMI, which is a known risk factor for TB, and high BMI or other obesity-related indicators, including waist circumference and hip circumference, and the risk of TB is controversial.

In practice, randomized controlled studies are subject to some bias and a variety of other confounding factors, resulting in less reliable results [20]. Mendelian randomization (MR) analysis utilizes single nucleotide polymorphisms (SNPs) in whole genome sequencing data as instrumental variables (IVs) [21], thereby enabling assessment of the causal effect of exposure factors on outcome(s) [22]. Moreover, MR analysis has the advantage of relying on the randomization of alleles during meiosis to eliminate the confounding effect of instrumental variables [23]. MR analysis relies on requisite basic assumptions, the first of which is that IVs should be highly correlated with exposure factors. Secondly, these IVs should not act on the outcome through confounding factors to ensure that they are independent. Finally, it should be ensured that the study affects the outcome through exposure factors only [24, 25]. In addition, high-throughput technology using genome-wide association studies (GWAS) is more conducive to large-scale experimental studies.

Based on the above, the present study obtained sequencing data for obesity-related indicators, including BMI, waist circumference, hip circumference, and waist-to-hip ratio, from a GWAS database and used MR analysis to explore their causal effects on TB. Using this method, the present study provides a valid causal assessment of risk factors for TB based on large-sample data.

## Material and methods

### IV acquisition and screening process

The overall design of this study is illustrated in the flow diagram presented in Fig 1. In this study, genetic variation data for obesity-related indicators, including BMI [26], waist

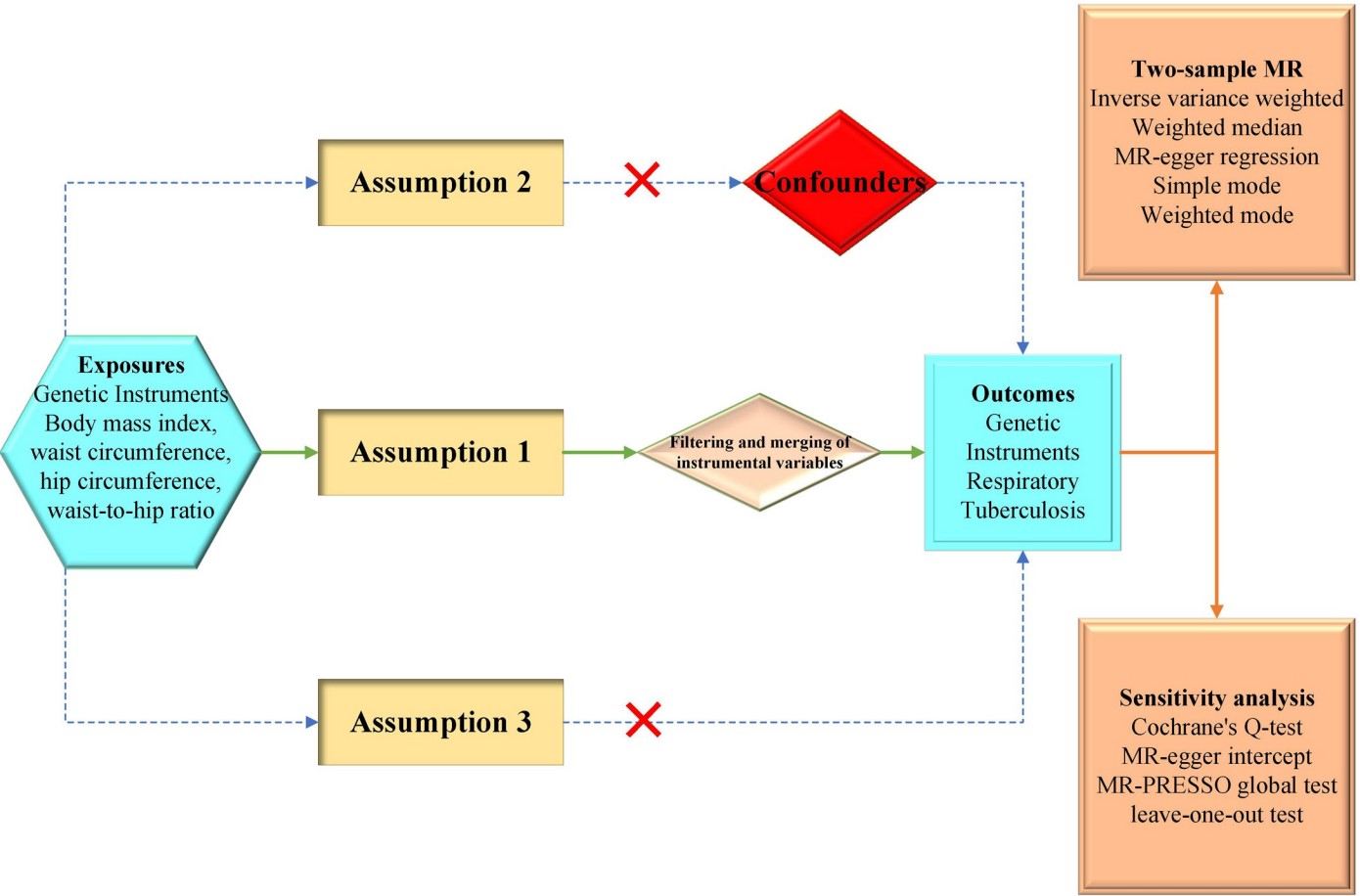

**Fig 1. Flow chart of the experimental design of this study.** SNPs for obesity-related indicators were identified as genetic instrumental variables. Assumption 1: The genetic variations are strongly associated with exposure; Assumption 2: The genetic variations are not associated with either known or unknown confounders; Assumption 3: SNPs should influence risk of the outcome through the exposure, not through other pathways.

circumference [27], hip circumference [27], waist-to-hip ratio [27], were obtained from the Genetic Investigation of Anthropometric Traits (GIANT) Consortium. The genetic variation data associated with TB was obtained from the Finngen biobank. Based on our comprehensive literature search, we have identified three TB-related factors as potential confounding variables that could impact the genuine causal relationship between obesity-related indicators and TB: smoking [28], type 2 diabetes [29], and educational attainment [30]. All the data were sourced from distinct cohorts of European ancestry, with exposures and outcomes stemming from different study, devoid of overlap.

First, SNPs that were strongly associated with exposure factors were screened. The screening criteria were $p < 5 \times 10^{-8}$, distance of linkage disequilibrium (LD) > 10000 kb and $r^2 <$ 0.001. In our screening criteria, kb represents the length of the region between the LDs considered, while $r^2 = 1$ represents a complete LD relationship between two SNPs, and $r^2 = 0$ represents a complete linkage equilibrium. Next, to ensure that the screened SNPs were free from

potential confounders, all SNPs included in the study were examined using PhenoScanner V2 (http://www.phenoscanner.medschl.-cam.ac.uk/). Factors associated with a high risk for TB, including smoking, type 2 diabetes, and educational attainment, were excluded. Finally, to ensure the strong predictive power of the screened SNPs, the corresponding F-statistic values were calculated. F = (N-k-1)/k × $R^2$/(1-$R^2$), where N represents the sample size in the exposed data, k is the number of IVs, and $R^2$ is the coefficient of determination.

## Filtering of outcome data

In this study, the finn-b-TBC_RESP dataset was selected from the GWAS database as the outcome data. This dataset contained 849 TB patients as well as 217632 control samples, with a total of 16380466 SNPs detected. In addition, the SNPs of the IVs as exposure factors were intersected with the outcome data and used as the final SNPs for performing MR analysis. The ethnicity of the subjects from whom the data were sourced was European, and a total of 16,380,466 SNPs from 218,481 subjects were tested.

## Validation of the causal effect of exposure and outcome data

To explore the causal effects of obesity-related indicators on TB, six tests commonly used in two-sample MR analysis, including Inverse variance weighted (IVW), weighted median, MR-Egger regression, weighted mode, simple mode, and Mendelian Randomization Pleiotropy RESidual Sum and Outlier (MR-PRESSO) were applied. These methods provide a multifaceted analysis by detecting SNPs as a whole and individual SNPs from different perspectives, respectively, among which the IVW method is currently considered to have a more comprehensive validation efficacy [31, 32]. In this study, six tests were calculated for the MR analysis of each exposure factor, and the IVW test results were mainly used as the main result to determine the causal relationship. Given that our study evaluates four null hypotheses, we have employed a Bonferroni-corrected Type I error rate of αBonf = 0.0125(0.05/4 = 0.0125) to address the issue of multiple testing.

## Sensitivity test of two-sample MR

The bias introduced by pleiotropy in IVs may affect the outcome estimates, therefore, we used the following analyses to detect and address possible pleiotropy. We tested the heterogeneity among SNPs in the IVW model by Cochran Q heterogeneity test. Cochran Q's p-value > 0.05 and I2 index < 0.25 was considered as low-level heterogeneity. Secondly, egger regression intercepts and the MR-PRESSO global test were used to detect the presence of horizontal pleiotropy in the included IVs. In order to minimize the pleiotropy from smoking, type 2 diabetes, and educational attainment, we conducted a multivariable MR analysis with adjustments for genetically predicted of these variables. Finally, each SNP was examined using the leave-one-out method and to evaluate the results of MR again. The leave-one-out method involves gradually eliminating each SNP, calculating the meta-effects of the remaining SNPs, and observing whether changes occur after eliminating each SNP [33].

## Statistical analysis

Statistical analysis and mapping were performed using R 4.2.0 (R Foundation for Statistical Computing, Vienna, Austria) and MR analysis was performed using the "TwoSampleMR" and "MR-PRESSO" R packages. Thereafter, this study mainly determined the causal effect of TB exposure on TB using the IVW method and combined the consistency of other tests.

# Results

## Characterization of IVs

Using qualifying filtering, representative SNPs were selected as IVs for data analysis. During the screening process, palindromic sequences and SNPs associated with strong risk factors for TB, including rs10784502, rs12656497, rs2371767, rs3786897, rs459193, rs6905288, rs7759938, rs780159, rs806794, rs849140, were excluded. Ultimately, 30 SNPs for BMI, 55 for waist circumference, 75 for hip circumference, and 31 for waist-to-hip ratio were selected as IVs (Table 1). Details of these IVs are summarized in the S1–S4 Tables. In addition, to ensure the reliability of these IVs, the F-statistic was calculated separately and yielded a value > 10, demonstrating the credibility of the IVs selected for this study.

## Heterogeneity and pleiotropy tests

For the heterogeneity test, the p-values of both IVW and MR-Egger tests were > 0.05, indicating that these IVs were not heterogeneous (Table 2). In addition, the Q values of Cochrane's Q test reflected good test effects (27.25–51.17). Furthermore, the p-values for MR-Egger intercept and MR-PRESSO global test were > 0.05 for the pleiotropy test, indicating that the SNPs included had no pleiotropy and no abnormal outliers were found for the SNPS selected in this study (Table 3). Simultaneously, in the multivariable MR analysis adjusting for genetically predicted potential confounding factors, the results remained steadfast (S5–S8 Tables).

## Causal effect(s) of obesity-related indicators on TB

In this study, the causal effect of obesity-related indicators on TB was analyzed using six common analytical methods in a two-sample MR. The results of this analysis are summarized in Table 3. Waist circumference achieved statistical significance according to two analysis methods—IVW (p = 0.011, odds ratio (OR) = 2.13, 95% confidence interval (CI) = 1.19–3.80) and MR-PRESSO suggests no heterogeneity of results. (Fig 2A). However, hip circumference did not meet the corrected p-value according to four analysis methods—IVW (p = 0.03, OR = 1.66, 95% CI = 1.03–2.66), the results of weighted median, MR Egger and MR-PRESSO suggest no heterogeneity and validate a similar trend effect to the IVW test (Fig 2C). The distribution of IVs for waist circumference were relatively uniform in the funnel plot (Fig 2B). Results of this analysis indicated that waist circumference was likely to be a risk factor for TB and that there was a positive causal relationship: more specifically, the greater the waist circumference, the greater the probability of developing TB. Moreover, according to the results of analysis, BMI and waist-to-hip ratio were not found to have a causal relationship with TB (S1 Fig).

**Table 1. Characteristics of exposures' datasets.**

| Exposures | Consortium | Sample | nSNP | nIVs | F-statistic |
|---|---|---|---|---|---|
| BMI | GIANT | 152893 | 2477659 | 30 | 250.32 |
| Waist circumference | GIANT | 245746 | 2547573 | 55 | 63.19 |
| Hip circumference | GIANT | 225487 | 2542663 | 75 | 75.44 |
| Waist-to-hip ratio | GIANT | 224459 | 2562516 | 31 | 55.99 |

BMI: body mass index; SNP: single-nucleotide polymorphism; nSNP: number of SNPs; IVs: Instrumental variable; nIVs: number of IVs.

**Table 2. The estimations of heterogeneity and horizontal pleiotropy for MR results.**

| Exposures | Inverse variance weighted | | MR-Egger | | | | MR-PRESSO |
|---|---|---|---|---|---|---|---|
| | Q-statistic | p | Q-statistic | p | Egger intercept | p | p for global test |
| BMI | 27.28 | 0.56 | 27.25 | 0.50 | 4.19e-03(-0.05–0.06) | 0.88 | 0.24 |
| Waist circumference | 44.26 | 0.77 | 43.70 | 0.76 | 0.02(-0.04–0.09) | 0.46 | 0.80 |
| Hip circumference | 51.17 | 0.78 | 46.46 | 0.88 | -0.05(-0.09–1.26e-03) | 0.06 | 0.87 |
| Waist-to-hip ratio | 39.6 | 0.07 | 39.4 | 0.06 | 0.02(-0.09–0.13) | 0.65 | 0.08 |

## Sensitivity analysis

Finally, analysis using the "leave-one-out method" was performed to test whether the included IVs had data stability, by excluding individual SNPs in a stepwise manner (Fig 3). According to the results of the leave-one-out method, the results of the overall effect of the remaining method are shown in the red line at the bottom of the Fig 3. The red line of the overall effect is on the side greater than zero, indicating that waist and hip circumference are both risk factors for TB (Fig 3A and 3B). In contrast, the SNPs for BMI and waist-to-hip ratio demonstrated partial volatility and lacked sufficient evidence to support their causal effect on TB (Fig 3C and 3D).

## Calculation of power

The power corresponding to waist circumference in this study was calculated using an online tool (https://shiny.cnsgenomics.com/mRnd/). The relevant parameters are set as follows: type-I error rate is 0.05, $R^2$ is 0.0002, based on the overall sample size, the final power we get is 84%. This result suggest that this study has sufficient power to explore the causal relationship between waist circumference and TB.

## Discussion

The present study explored the causal relationship between obesity-related indicators and TB using two-sample MR analysis. We used the results of IVW as a criterion for judging the causal relationship between exposure and outcome [34], while the other methods assessed the validity of the exclusion assumption [35]. According to the analysis, all six tests for waist circumference revealed positive linear estimates of effect, suggesting that this exposure factor is likely to be potential risk factors for TB, and the greater this indicator, the higher the risk for developing TB. On the one hand, in the leave-one-out analysis, we found that IVs with waist

**Table 3. Two-sample Mendelian randomization estimations showing the effects of obesity-related indicators on the risk of respiratory tuberculosis.**

| Exposures | Inverse variance weighted | | Weighted median | | MR Egger | |
|---|---|---|---|---|---|---|
| | OR (95%CI) | p | OR (95%CI) | p | OR (95%CI) | p |
| BMI | 0.64(0.37–1.11) | 0.11 | 0.73(0.30–1.76) | 0.48 | 0.58(0.13–2.55) | 0.47 |
| Waist circumference | 2.13(1.19–3.80) | **0.011** | 2.56(1.13–5.83) | 0.03 | 0.87(0.08–9.58) | 0.91 |
| Hip circumference | 1.66(1.03–2.66) | 0.03 | 2.25(1.15–4.40) | 0.02 | 13.51(1.91–95.42) | 0.01 |
| Waist-to-hip ratio | 0.71(0.27–1.83) | 0.48 | 0.80(0.25–2.56) | 0.71 | 0.33(4.25e-03-25.67) | 0.62 |
| Exposures | Simple mode | | Weighted mode | | MR-PRESSO | |
| | OR (95%CI) | p | OR (95%CI) | p | OR (95%CI) | p |
| BMI | 4.46(0.74–26.82) | 0.57 | 2.19(0.48–9.95) | 0.63 | 0.64(0.38–1.09) | 0.11 |
| Waist circumference | 4.06(0.65–25.51) | 0.14 | 2.83(0.51–15.66) | 0.24 | 2.11(1.25–3.55) | 6.63e-03 |
| Hip circumference | 2.93(0.66–13.07) | 0.16 | 3.12(0.88–11.07) | 0.08 | 1.50(1.02–2.19) | 0.04 |
| Waist-to-hip ratio | 1.93(0.20–18.87) | 0.58 | 0.83(0.14–4.90) | 0.84 | 0.81(0.33–2.00) | 0.08 |

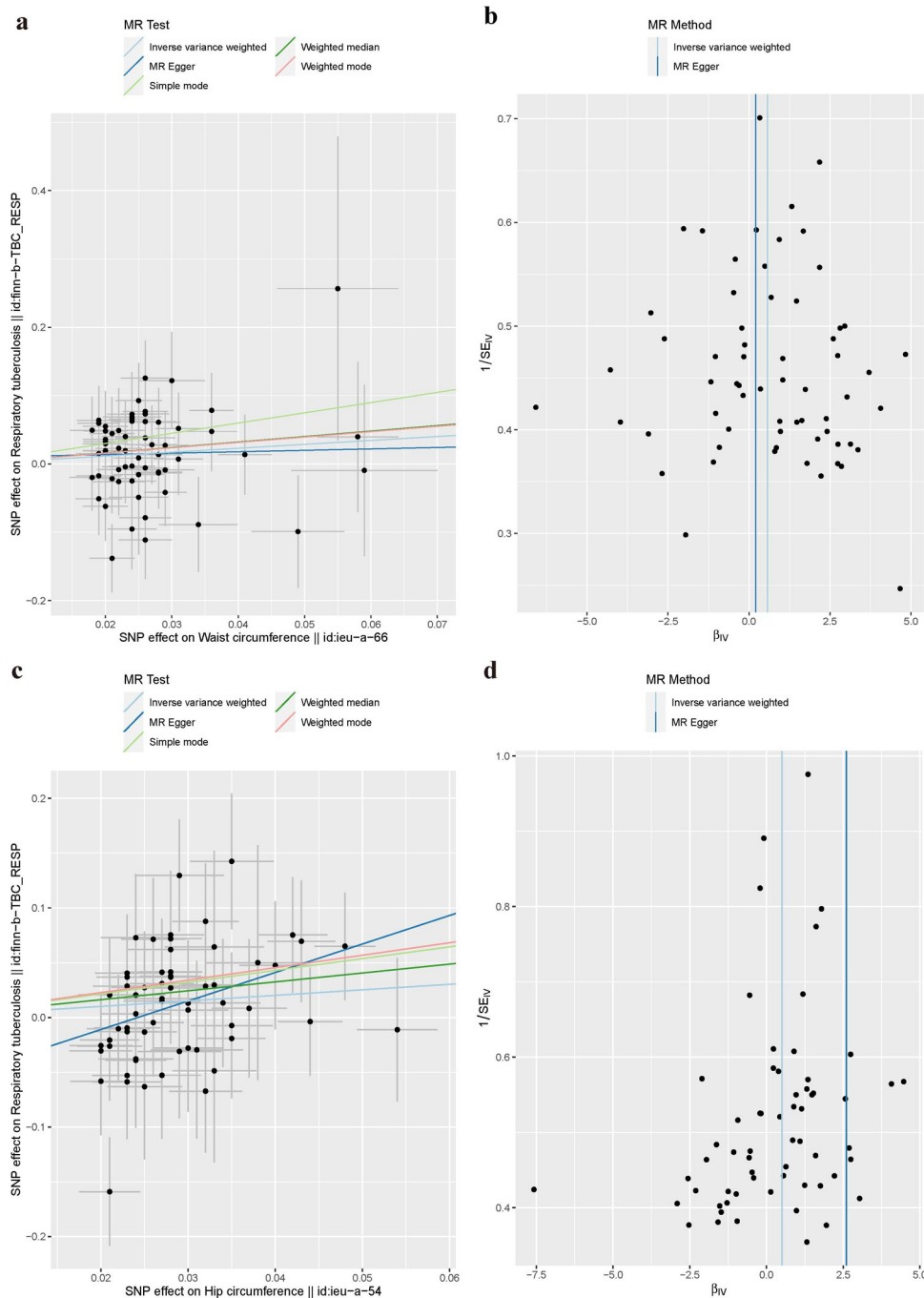

**Fig 2.** (a, c): Scatter plot of SNPs associated with waist circumference and risk of TB. The plot shows the SNP effects on waist circumference (x-axis, SD units) as well as TB (y-axis, OR) with 95% CI. The MR regression slopes of the lines represent the causal estimates using five approaches (IVW, MR-Egger, weighted median, simple mode, and weighted mode; (b, d): Funnel diagram corresponding to waist and hip circumference.

circumference had more stable outcomes, suggesting a strong causal effect of waist circumference on TB. On the other hand, in the leave-one-out test, we found that there are some IVs that affect the overall effect, which may be due to the more complex risk factors of TB and the presence of some potential influences. This leads to the results of IVW suggesting a significant

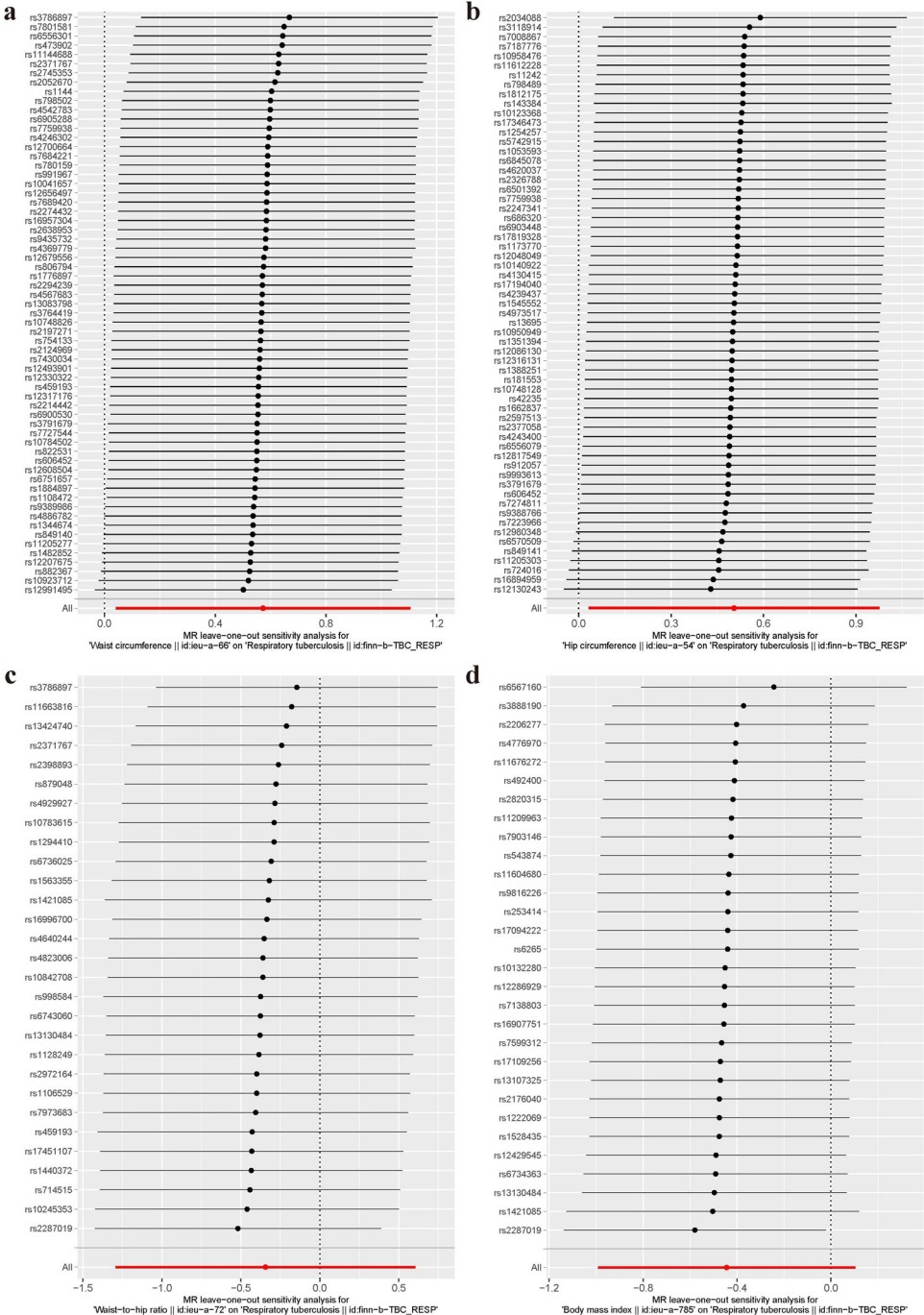

**Fig 3. Leave-one-out plots for causality analysis of waist circumference, hip circumference, waist-to-hip ratio and body mass index on TB.**

causal effect, but the results of other complementary tests such as MR egger, Simple and Weighted mode have larger p-values. Regarding the three indicators, BMI, hip circumference and waist-to-hip ratio, no potential effect on TB was found in this study. It may be due to lack of statistical power or sufficient IVs to identify a true causal relationship.

Waist circumference is not only common indicator of visceral fat and leg fat accumulation, but also reflect body fat distribution and overall metabolic level [36–39]. It has been established that abdominal obesity can lead to many diseases. Studies have shown that abdominal obesity and insulin resistance are closely related and ultimately lead to an increased risk for cardiovascular disease and even the development of type 2 diabetes and atherosclerosis [40–43]. A recent MR study reported a causal relationship between waist circumference and type 2 diabetes, and type 2 diabetes also leads to an increased risk of TB [44]. One of the potential regulatory mechanisms may be related to the involvement of some genes in the metabolism and regulation of fat, thereby altering its distribution in the body [27, 45]. In addition, some genes are involved in the regulation of insulin sensitivity through signaling pathways, such as the tyrosine kinase signaling pathway, which acts on the insulin receptor leading to its phosphorylation, further leading to the inhibition of insulin receptor signaling, and thus, to a decrease in sensitivity [46–48]. These regulatory mechanisms can lead to cellular aging [49] and impaired mitochondrial function [50], as well as impaired immune functions in the body [51, 52]. Relevant studies have demonstrated a strong link between TB and impaired immune response [53]. One study elucidated another potential regulatory mechanism of the intestine–lung axis [54], suggesting that obesity can lead to dysbiosis of the intestinal flora and dysregulation of the intestinal microenvironment, thereby increasing susceptibility to TB and decreasing tolerance to lung injury. Although some studies have reported that TB is strongly associated with impaired immune response and that the risk for latent TB recurrence is extremely low in immunocompetent patients, the risk for recurrence is, in fact, significantly elevated in this patient population [55].

Several relevant clinical cohort studies have investigated the relationship between obesity and TB; however, the conclusions drawn have varied widely. A prospective cohort study from China noted that participants with a BMI > 24 kg/m$^2$ and diagnosed with diabetes had a similar risk for developing TB as those who were overweight but did not have diabetes [56]. A systematic review suggests that BMI and risk of TB show an inverse relationship and that this inverse relationship applies from underweight to obese [57]. Moreover, a cross-sectional study from the United States reported a negative association between BMI and latent TB, and reduced BMI could be considered a risk factor for latent TB [58]. Another recent study used artificial neural networks to predict the effect of obesity on TB, which was found to be a risk indicator for TB [59]. Clinical studies of waist and hip circumference on the risk of developing TB are currently lacking, focusing mainly on the analysis of BMI. In the present study, smoking, diabetes, and education failed to mediate the association between waist circumference and TB. This result suggests a potential pathogenic mechanism of waist circumference in TB. An increase in waist circumference may indicate disturbances in the distribution and metabolism of body fat, which may affect the body's immune defenses. Many people work and live a sedentary lifestyle, which leads to the accumulation of fat in the waist and hips, resulting in abdominal obesity [60]. Excessive fat metabolites in the body will cause endocrine disorders in the body [61], and may even affect the immune function of the body, thus decreasing immunity and thus increasing the risk of developing TB. To our knowledge, the present study is the first to explore the causal effect of obesity-related indicators on TB by means of large-sample data analysis, compensating for the small sample sizes and presence of confounding factors and bias in previous clinical cohort studies.

However, this study had some limitations. First, there were limitations in the data; the various obesity-related indicators were only overall data and are not disaggregated in detail. Second, this study identified the relationship between waist circumference and TB as linear, whereas some other studies have reported that low BMI is also a risk factor for TB [62]; as such, further research is required. At the same time, the data of F-statistic value in the study of

large sample data analysis will be very large, but the corresponding validation effectiveness will be reduced. In addition, some excessive confidence intervals were obtained in the MR analysis, and the sample size and IVs need to be expanded in the future. Socioeconomic status was also an important risk factor for TB when mediation analyses were performed; however, a corresponding dataset could not be found in the GWAS database, and this important risk factor needs to be further analyzed. Finally, the data sample selected for this study was mainly European; as such, there is a need to expand the population to improve the generalizability of the experimental results.

## Conclusion

In summary, we determined a causal relationship between waist circumference and TB using two-sample MR analysis, which suggested that waist circumference is likely to be potential risk factors. The present study enriches the study of risk factors for the development of TB, and the potential mechanisms of action need to be further explored.

## Supporting information

**S1 Fig.** (a, c): Scatter and funnel plots of body mass index. (c, d): Scatter and funnel plot of waist-hip ratio.
(PDF)

**S1 Table. Harmonized dataset of Mendelian randomization for the effect of body mass index on respiratory tuberculosis.**
(DOCX)

**S2 Table. Harmonized dataset of Mendelian randomization for the effect of waist-to-hip ratio on respiratory tuberculosis.**
(DOCX)

**S3 Table. Harmonized dataset of Mendelian randomization for the effect of hip circumference on respiratory tuberculosis.**
(DOCX)

**S4 Table. Harmonized dataset of Mendelian randomization for the effect of waist circumference on respiratory tuberculosis.**
(DOCX)

**S5 Table. Multivariate MR analysis of BMI and smoking, type 2 diabetes and educational attainment.**
(DOCX)

**S6 Table. Multivariate MR analysis of waist circumference and smoking, type 2 diabetes and educational attainment.**
(DOCX)

**S7 Table. Multivariate MR analysis of hip circumference and smoking, type 2 diabetes and educational attainment.**
(DOCX)

**S8 Table. Multivariate MR analysis of waist-to-hip ratio and smoking, type 2 diabetes and educational attainment.**
(DOCX)

## Author Contributions

**Methodology:** Nuannuan Cai, Weiyan Luo, Lili Ding, Lijin Chen.

**Project administration:** Nuannuan Cai, Yuanjiang Huang.

**Writing – original draft:** Nuannuan Cai, Weiyan Luo.

**Writing – review & editing:** Lili Ding, Lijin Chen, Yuanjiang Huang.

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
