## [Decision Letter · Decision Letter 0]

2 May 2023

PONE-D-23-09863Obesity-related indicators and tuberculosis a Mendelian randomization studyPLOS ONE

Dear Dr. Huang,

Thank you for submitting your manuscript to PLOS ONE. After careful consideration, we feel that it has merit but does not fully meet PLOS ONE’s publication criteria as it currently stands. Therefore, we invite you to submit a revised version of the manuscript that addresses the points raised during the review process.

We look forward to receiving your revised manuscript.

Kind regards,

Chunyu Liu, PhD

Academic Editor

PLOS ONE

Journal Requirements:

   "This study was supported by the impact of 4C continuity of care model on non-invasive ventilator use in patients with chronic obstructive pulmonary disease (Project number: 2101320712A2012). "

Reviewers' comments:

Reviewer's Responses to Questions

**Comments to the Author**

1. Is the manuscript technically sound, and do the data support the conclusions?

Reviewer #1: Yes

Reviewer #2: Partly

2. Has the statistical analysis been performed appropriately and rigorously? 

Reviewer #1: Yes

Reviewer #2: Yes

3. Have the authors made all data underlying the findings in their manuscript fully available?

Reviewer #1: Yes

Reviewer #2: Yes

4. Is the manuscript presented in an intelligible fashion and written in standard English?

Reviewer #1: Yes

Reviewer #2: Yes

5. Review Comments to the Author

Reviewer #1: 1. Please all the obesity indicators in the abstract to facilitate readers.

2. Line 43, is M. tuberculosis the abbreviation of Mycobacterium tuberculosis? Please specify.

3. Please polish the language and transition between paragraphs. For example, Line 52, it seems too sudden jumping from the first paragraph to the second paragraph; Line 59, two “however”s in two sentences; etc.

4. Line 72, the second assumption of MR seems not correct. There is no assumption that IVs should be uncorrelated with each other. And the authors mentioned these three assumptions again in the IV selection part, and the second assumption is a different one there.

5. Line 85, the citation of GIANT BMI GWAS is from 2013 paper, but as far as I know, the Locke et al. 2015 is the GIANT BMI GWAS paper that corresponds to the GWAS summary stats on GIANT website (https://portals.broadinstitute.org/collaboration/giant/index.php/GIANT_consortium_data_files#GWAS_Anthropometric_2015_BMI_Summary_Statistics). Can you check which GWAS summary stats was actually used in the paper?

6. Line 86, is waist-hip-ratio the actual exposure used? Or waist-hip-ratio adjusted for BMI?

7. Line 108, what is genetic data point?

8. The authors first present the main analyses using six methods, then go to Heterogeneity and horizontal multiplicity tests and leave-one-out analyses as sensitivity analyses. But from my understanding, the logic should be first check heterogeneity and pleiotropy, if there is any IVs with pleiotropy (can be identified by MR-PRESSO individual test), then remove those problematic IVs and perform the primary MR analyses. So I would suggest restructure the results section to check heterogeneity and pleiotropy -> primary MR analyses -> leave one out as sensitivity analysis.

9. In Table 2, sometimes results are different between different methods, for example, for waist circumference, it is significant in IVW but not significant in MR egger. May consider discuss those discrepancies in the discussion section.

10. We need higher resolution for all the figures.

Reviewer #2: Summary:

This is a two-sample Mendelian randomization study making use of publicly available genetic association summary statistics to investigate causal relationships between anthropometric traits and tuberculosis risk. The authors identify consistent evidence for a causal relationship between waist-circumference and TB, and possible evidence for a causal relationship between hip-circumference and TB.

The statistical design and reporting of the study is overall done well. While I am not aware of any similar studies, I am not thoroughly acquainted with the applied Mendelian randomization literature in the tuberculosis space.

Major Remarks:

1. The reporting of causal effect estimates from Mendelian randomization should be chosen from a method identified as most appropriate for the data. In this case, I would recommend reporting the IVW estimate and P-value as the primary finding unless other methods suggest a presence of horizontal pleiotropy or sensitivity analyses show evidence of heterogeneity. This is consistent with the author’s method section suggesting that the IVW method is preferred and will be the primary focus. (Lines 117-118 & Line 133-134) Therefore, the results section should be framed as reporting the IVW estimate foremost unless Table 3 shows substantial evidence of heterogeneity for the outcome or the pleiotropy robust methods (median, PRESSO) produce substantially different effect sizes.

Based on this, for the waist circumference outcome I would recommend highlighting the effect size and P-Value from the IVW method primarily, and commenting that no heterogeneity was observed and that pleiotropy robust methods like weighted median and PRESSO provided similar effect sizes.

2. The hip-circumference results differ between those reported in text on lines 166-170 versus Table 2. Please clarify which set of results are the actual results for the hip-circumference outcome. If the table results are accurate, please explain why the MR-PRESSO method alone should categorize hip-circumference as a significant finding while no evidence of heterogeneity or pleiotropy is observed in Table 3 along with no difference in effect sizes for the IVW (1.49) and robust methods (Median=1.50, PRESSO = 1.50). Additionally, the substantial increase in MR-Egger effect size should be explained.

3. For those outcomes being reported as significantly causally associated with TB, it should be explained why the Mode & Weighted Mode methods produce substantially larger effect sizes and P-Values than the other methods.

 

Minor Remarks:

1. It should be made clear that the IVW method relies on the assumptions of MR for unbiased estimation, while the other five methods (medians, modes, & PRESSO) all attempt to correct for violations of the exclusion assumption.

2. More detail should be provided in the main text about the resources used for instrument selection and the corresponding association studies for the exposures and outcome.

3. The leave-one-out analysis is not sufficiently explained in the methods and it is hard for me to interpret the results without the supplementary figures.

Minor line-by-line suggestions:

Line 60: The last sentence of the second paragraph in the introduction seems contradictory. The authors state that low BMI is a known risk factor for TB, the subsequently state that BMI TB associations are controversial. Some clarification or expanded explanation here may be useful.

Line 67: MR is not necessarily a whole-genome sequence (WGS) technique.

Line 70: High correlation with the exposure does not ensure that no other confounding factors are present. MR relies on the randomization of alleles during meiosis to remove instrumental variables from confounding effects.

Line 77: Sequencing data… Are these studies WGS?

Line 85: More details provided on the exposure GWAS should be provided.

Line 90: More details on the LD exclusion should be provided. Was pairwise LD computed for each instrument, and if so, what resource was used.

Line 95: Provide details on F-Statistic computation. Additionally, given the large sample sizes, F-statistics may not be very informative.

Line 105: Sequenced… Was the data obtained from WGS study? More details on the outcome GWAS would be helpful.

Line 106: What is meant by the point at which IVs intersected with the exposure data? And how was it incorporated into the MR methods?

Line 118: Citation for IVW validation efficacy?

Line 119: A justification for 0.05 as a significance threshold may be helpful given the 6 tests for each outcome.

Line 145-147: With such large sample sizes, large F-statistics are not surprising and may not give much insight into instrument strength.

Line 173: positive causal relationships

Line 176: Non-significant findings do not necessarily suggest a lack of a causal relationship but may be due to lack of statistical power or sufficient instruments to identify a true causal relationship.

Line 202: indicators and TB using two-sample mendelian randomization.

Line 202-205: These MR methods provide linear estimates of effect, not correlations.

6. PLOS authors have the option to publish the peer review history of their article (what does this mean?). If published, this will include your full peer review and any attached files.

Reviewer #1: No

Reviewer #2: **Yes: **Daniel DiCorpo

---

## [Author Response · Author response to Decision Letter 0]

11 Jun 2023

Dear PhD. Chunyu Liu, 

Thank you for your letter and for the reviewers’ comments concerning our manuscript entitled “Obesity-related indicators and tuberculosis a Mendelian randomization study” (ID: PONE-D-23-09863). Those comments are all valuable and very helpful for revising and improving our paper, as well as the important guiding significance to our researches. We have studied comments carefully and have made correction which we hope meet with approval. In tracking the revised manuscript, we yellow the revised and supplemented parts. 

We have fully addressed each concern and hope that this revised manuscript is now acceptable. The main corrections in the paper and the responds to the reviewer’s comments are as following:

Responds to the Journal requirements: 

1. Response to comment: Please ensure that your manuscript meets PLOS ONE's style requirements, including those for file naming.

Response: Thank you for the reminder. We have named the submitted files as requested and the format of the manuscript has been revised as requested.

2. Response to comment: Thank you for stating the following financial disclosure:

"This study was supported by the impact of 4C continuity of care model on non-invasive ventilator use in patients with chronic obstructive pulmonary disease (Project number: 2101320712A2012). "

Response: The funders had no role in study design, data collection and analysis, decision to publish, or preparation of the manuscript.

Responds to the Reviewer’s comments: 

Reviewer #1:

1. Response to comment: Please all the obesity indicators in the abstract to facilitate readers.

Response: Thank you for your suggestions for this article. We have added obesity-related indicators in the second paragraph of the introduction section. The additions are as follows: In clinical practice, obesity-related indicators include body mass index (BMI), waist circumference, BMI-adjusted waist-to-hip ratio, hip circumference, arm circumference, blood lipid level, fat mass and low lean body mass. (Page. 3, Line. 56-59)

2. Response to comment: Line 43, is M. tuberculosis the abbreviation of Mycobacterium tuberculosis? Please specify.

Response: We apologize for the inconvenience of reading this article due to an oversight on our part. We have made a correction to add the full name of Mycobacterium tuberculosis. (Page. 3, Line. 42)

3. Response to comment: Please polish the language and transition between paragraphs. For example, Line 52, it seems too sudden jumping from the first paragraph to the second paragraph; Line 59, two “however”s in two sentences; etc. 

Response: Thanks to your valuable comments, we have made changes to the overall language of the article. As you pointed out in the paragraph-to-paragraph connection and sentence-to-sentence conjunction, we made changes. 

4. Response to comment: Line 72, the second assumption of MR seems not correct. There is no assumption that IVs should be uncorrelated with each other. And the authors mentioned these three assumptions again in the IV selection part, and the second assumption is a different one there.

Response: Your suggestions are very valuable. We apologize for the lack of clarity due to an oversight on our part. The second assumption of Mendelian randomization should be that IVs are independent of confounding factors and exhibit independence of instrumental variables. Therefore, we have made the following changes in the text: Secondly, these IVs should not act on the outcome through confounding factors to ensure that they are independent. (Page. 4, Line. 75-76)

5. Response to comment: Line 85, the citation of GIANT BMI GWAS is from 2013 paper, but as far as I know, the Locke et al. 2015 is the GIANT BMI GWAS paper that corresponds to the GWAS summary stats on GIANT website (https://portals.broadinstitute.org/collaboration/giant/index.php/GIANT_consortium_data_files#GWAS_Anthropometric_2015_BMI_Summary_Statistics). Can you check which GWAS summary stats was actually used in the paper?

Response: Thank you very much for the tip, and we are sorry that we have not been very strict in correcting the references. As you said, we made the change to cite the correct reference, and we cited Genetic studies of body mass index yield new insights for obesity biology (PMID: 25673413). (Page. 5, Line. 88) 

6. Response to comment: Line 86, is waist-hip-ratio the actual exposure used? Or waist-hip-ratio adjusted for BMI? 

Response: Thank you for your comments. The waist-to-hip ratio selected for this study is from the Dataset: ieu-a-72, and the data provided are for the waist-to-hip ratio, not for the BMI adjusted.

7. Response to comment: Line 108, what is genetic data point?

Response: We are very sorry that we did not express ourselves clearly. We have changed genetic data point to SNPs. (Page. 6, Line. 124)

8. Response to comment: The authors first present the main analyses using six methods, then go to Heterogeneity and horizontal multiplicity tests and leave-one-out analyses as sensitivity analyses. But from my understanding, the logic should be first check heterogeneity and pleiotropy, if there is any IVs with pleiotropy (can be identified by MR-PRESSO individual test), then remove those problematic IVs and perform the primary MR analyses. So I would suggest restructure the results section to check heterogeneity and pleiotropy -> primary MR analyses -> leave one out as sensitivity analysis. 

Response: Thank you for your suggestion, and we think it is very valuable. Based on your valuable suggestions, we have adjusted the order of the results section, as well as the order of Table 2 and Table 3. The result section is modified as follows: the second part of the result is modified to Heterogeneity and horizontal multiplicity tests, and the fourth part is added to Sensitivity analysis. (Page. 9, Line. 182-188; Page. 12-13, Line. 214-222)

9. Response to comment: In Table 2, sometimes results are different between different methods, for example, for waist circumference, it is significant in IVW but not significant in MR egger. May consider discuss those discrepancies in the discussion section. 

Response: Your advice is very valuable. We have added in the discussion section to explain this situation. The additions are as follows: On the other hand, in the leave-one-out test, we found that there are some IVs that affect the overall effect, which may be due to the more complex risk factors of TB and the presence of some potential influences. This leads to the results of IVW suggesting a significant causal effect, but the results of other complementary tests such as MR egger, Simple and Weighted mode have larger p-values. (Page. 13-14, Line. 236-240)

10. Response to comment: We need higher resolution for all the figures. 

Response: We are very sorry that the clarity of our previous pictures was not adjusted properly. We have adjusted all the images according to your valuable suggestions and modified the clarity of all the figures. (Figure 1, 2, 3 and Supplementary Figure 1)

Reviewer #2: 

This is a two-sample Mendelian randomization study making use of publicly available genetic association summary statistics to investigate causal relationships between anthropometric traits and tuberculosis risk. The authors identify consistent evidence for a causal relationship between waist-circumference and TB, and possible evidence for a causal relationship between hip-circumference and TB.

The statistical design and reporting of the study is overall done well. While I am not aware of any similar studies, I am not thoroughly acquainted with the applied Mendelian randomization literature in the tuberculosis space.

Response to comment: Thank you very much for reviewing our manuscript, your suggestions are all very valuable. In particular, thank you for your careful review of our manuscript and for guiding us to a deeper understanding of Mendelian randomization studies. Thank you again for your hard work. 

Major Remarks:

1. Response to comment: The reporting of causal effect estimates from Mendelian randomization should be chosen from a method identified as most appropriate for the data. In this case, I would recommend reporting the IVW estimate and P-value as the primary finding unless other methods suggest a presence of horizontal pleiotropy or sensitivity analyses show evidence of heterogeneity. This is consistent with the author’s method section suggesting that the IVW method is preferred and will be the primary focus. (Lines 117-118 & Line 133-134) Therefore, the results section should be framed as reporting the IVW estimate foremost unless Table 3 shows substantial evidence of heterogeneity for the outcome or the pleiotropy robust methods (median, PRESSO) produce substantially different effect sizes.

Based on this, for the waist circumference outcome I would recommend highlighting the effect size and P-Value from the IVW method primarily, and commenting that no heterogeneity was observed and that pleiotropy robust methods like weighted median and PRESSO provided similar effect sizes.

Response: Thank you very much for your advice, which is very professional. Based on your valuable suggestions we have redescribed in the results section, mainly for IVW, the rest of the test is supplementary to the description of the role. The modifications we have made are as follows: MR-PRESSO suggests no heterogeneity of results (Page. 11, Line. 198); the results of weighted median, MR Egger and MR-PRESSO suggest no heterogeneity and validate a similar trend effect to the IVW test (Page. 11, Line. 201-202)

2. Response to comment: The hip-circumference results differ between those reported in text on lines 166-170 versus Table 2. Please clarify which set of results are the actual results for the hip-circumference outcome. If the table results are accurate, please explain why the MR-PRESSO method alone should categorize hip-circumference as a significant finding while no evidence of heterogeneity or pleiotropy is observed in Table 3 along with no difference in effect sizes for the IVW (1.49) and robust methods (Median=1.50, PRESSO = 1.50). Additionally, the substantial increase in MR-Egger effect size should be explained.

Response: Thank you for your suggestions. We are very, very sorry for the problem in presenting the results for hip circumference in Table 3(we modify the original Table 2 to Table 3). The results presented in Table 3 are the results of our preliminary MR analysis, based on which we removed some of the SNPs related to confounding factors as described in the text, but due to an oversight there was an editing error in Table 3. We re-edited the data in Table 3, and the data depicted in the second part of the results in the text are correct. This problem is very serious and we apologize again. 

3. Response to comment: For those outcomes being reported as significantly causally associated with TB, it should be explained why the Mode & Weighted Mode methods produce substantially larger effect sizes and P-Values than the other methods.

Response: Your advice is very valuable. Based on the results we got, we have analyzed them and explained them in the discussion section. The additions are as follows: On the other hand, in the leave-one-out test, we found that there are some IVs that affect the overall effect, which may be due to the more complex risk factors of TB and the presence of some potential influences. This leads to the results of IVW suggesting a significant causal effect, but the results of other complementary tests such as MR egger, Simple and Weighted mode have larger p-values. (Page. 13-14, Line. 236-240)

Minor Remarks:

1. Response to comment: It should be made clear that the IVW method relies on the assumptions of MR for unbiased estimation, while the other five methods (medians, modes, & PRESSO) all attempt to correct for violations of the exclusion assumption.

Response: Thank you very much for your guidance. Based on your valuable suggestions we have added a description of the six methods of MR analysis examination in the manuscript in the discussion section. The additions are as follows: We used the results of IVW as a criterion for judging the causal relationship between exposure and outcome, which relied on unbiased estimates of the three assumptions of the MR analysis, distinguishing it from the other five tests, which mainly assessed the correction of the exclusion assumption. (Page. 13, Line. 227-230)

2. Response to comment: More detail should be provided in the main text about the resources used for instrument selection and the corresponding association studies for the exposures and outcome.

Response: Thank you for your valuable advice. We have added a description of exposure and outcomes and details of screening for IVs in the Materials and Methods section. The additions are as follows: 

(1) BMI (unit: kg/m^2) was selected from the ieu-a-785 dataset, containing 152893 samples, and the sequencing results of 2477659 SNPs. Waist circumference (unit: cm) was selected from the ieu-a-66 dataset, containing 245,756 samples, and the sequencing results of 254,7573 SNPs. Hip circumference (unit: cm) was selected from the ieu-a-54 dataset, containing 225487 samples, and the sequencing results of 2542663 SNPs. Waist-hip ratio was selected from the ieu-a-72 dataset, containing 224459 samples, and sequencing results of 2562516 SNPs. (Page. 5, Line. 90-97)

(2) The screening criteria were p < 5×10-8, distance of linkage disequilibrium (LD) > 10000 kb and r2 < 0.001. In our screening criteria, kb represents the length of the region between the LDs considered, while r2 = 1 represents a complete LD relationship between two SNPs, and r2 = 0 represents a complete linkage equilibrium. (Page. 5, Line. 98-102)

(3) In this study, the finn-b-TBC_RESP dataset was selected from the GWAS database as the outcome data. This dataset contained 849 TB patients as well as 217632 control samples, with a total of 16380466 SNPs detected. (Page. 6, Line. 119-121)

3. Response to comment: The leave-one-out analysis is not sufficiently explained in the methods and it is hard for me to interpret the results without the supplementary figures.

Response: Thank you for your valuable advice. We added an explanation of the leave-one-out method in paragraph Sensitivity test of two-sample MR of the Material and methods section and added a figure of the results of the leave-one-out method in Figure 3. The additions are as follows: The leave-one-out method involves gradually eliminating each SNP, calculating the meta-effects of the remaining SNPs, and observing whether changes occur after eliminating each SNP. (Page. 7, Line. 146-148; Figure 3) 

Minor line-by-line suggestions:

1. Response to comment: Line 60: The last sentence of the second paragraph in the introduction seems contradictory. The authors state that low BMI is a known risk factor for TB, the subsequently state that BMI TB associations are controversial. Some clarification or expanded explanation here may be useful.

Response: Your advice is very helpful. Based on your valuable suggestions, we have revised the description in the second paragraph of the introduction. Our intention was to express that previous studies have shown that low body mass index is a high risk factor for tuberculosis, but the risk of high body mass index for tuberculosis is controversial; therefore, we wanted to further investigate the effect of obesity on the risk of developing tuberculosis. 

 We have made the following changes in the manuscript: This is quite different from p0revious studies, which have shown that being underweight or having a low body mass index (BMI) is a risk factor for active TB. However, the evidence on the effect of obesity on tuberculosis is not well documented, and the association between low BMI, which is a known risk factor for TB, and high BMI or other obesity-related indicators, including waist circumference and hip circumference, and the risk of TB is controversial. (Page. 3-4, Line. 61-66)

2. Response to comments: Line 67: MR is not necessarily a whole-genome sequence (WGS) technique. 

Response: Thank you very much for your valuable advice and your advice is very helpful. Our description's inaccurate. In the manuscript we revised the following: Mendelian randomization (MR) analysis utilizes single nucleotide polymorphisms (SNPs) in whole genome sequencing data as instrumental variables (IVs). (Page. 4, Line. 68-70)

3. Response to comments: Line 70: High correlation with the exposure does not ensure that no other confounding factors are present. MR relies on the randomization of alleles during meiosis to remove instrumental variables from confounding effects.

Response: Thank you very much for pointing out our flaws. Based on your valuable comments, we have redescribed the hypothetical principle of Mendelian randomization. The modifications are as follows: Moreover, MR analysis has the advantage of relying on the randomization of alleles during meiosis to eliminate the confounding effect of instrumental variables. MR analysis relies on requisite basic assumptions, the first of which is that IVs should be highly correlated with exposure factors. Secondly, these IVs should not act on the outcome through confounding factors to ensure that they are independent. (Page. 4, Line. 71-76)

4. Response to comments: Line 77: Sequencing data… Are these studies WGS?

Response: Thanks for your comment. We would like to express that the GWAS database contains a wide range of populations with different characteristics and the corresponding whole genome sequencing data and that the database is useful for large scale population studies. We have modified the text as follows: In addition, the Genome-Wide Association Study (GWAS) database provides a large amount of whole-genome sequencing data for different characteristics of the population, which is more conducive to large-scale experimental studies. (Page. 4, Line. 89-92) 

5. Response to comments: Line 85: More details provided on the exposure GWAS should be provided. 

Response: Your advice is very important. Based on your suggestions, we have supplemented the description of the exposure data with the following additions: BMI (unit: kg/m^2) was selected from the ieu-a-785 dataset, containing 152893 samples, and the sequencing results of 2477659 SNPs. Waist circumference (unit: cm) was selected from the ieu-a-66 dataset, containing 245,756 samples, and the sequencing results of 254,7573 SNPs. Hip circumference (unit: cm) was selected from the ieu-a-54 dataset, containing 225487 samples, and the sequencing results of 2542663 SNPs. Waist-hip ratio was selected from the ieu-a-72 dataset, containing 224459 samples, and sequencing results of 2562516 SNPs. (Page. 5, Line. 90-97) 

6. Response to comment: Line 90: More details on the LD exclusion should be provided. Was pairwise LD computed for each instrument, and if so, what resource was used.

Response: Your suggestion is very helpful. The MR method to calculate LD is achieved by the adjustment of two parameters kd and r2 in the TwoSampleMR package. Therefore, we have described them in detail in the manuscript, with the following additions: The screening criteria were p < 5×10-8, distance of linkage disequilibrium (LD) > 10000 kb and r2 < 0.001. In our screening criteria, kb represents the length of the region between the LDs considered, while r2 = 1 represents a complete LD relationship between two SNPs, and r2 = 0 represents a complete linkage equilibrium. (Page. 5, Line. 98-102) 

7. Response to comment: Line 95: Provide details on F-Statistic computation. Additionally, given the large sample sizes, F-statistics may not be very informative.

Response: Your advice is very helpful. We supplemented the manuscript with the formula for the F-statistic with the following additions: F = (N-k-1)/k × R2/(1-R2), where N represents the sample size in the exposed data, k is the number of IVs, and R2 is the coefficient of determination. (Page. 5-6, Line. 107-109)

8. Response to comment: Line 105: Sequenced… Was the data obtained from WGS study? More details on the outcome GWAS would be helpful.

Response: Your reminder is very valuable and helpful to us. The SNPs for the endings we used were all from the GWAS database. Based on your suggestion, we have added to the description of the outcome data with the following modifications: In this study, the finn-b-TBC_RESP dataset was selected from the GWAS database as the outcome data. This dataset contained 849 TB patients as well as 217632 control samples, with a total of 16380466 SNPs detected. (Page. 6, Line. 119-123) 

9. Response to comment: Line 106: What is meant by the point at which IVs intersected with the exposure data? And how was it incorporated into the MR methods?

Response: Your correction is very meaningful. What we describe is to use the IVs of the exposed data to take the intersection in the outcomes data, and the SNPs after taking the intersection as the final IVs for MR analysis. We are sorry that our previous description was inaccurate. The modifications we made are as follows: In addition, the SNPs of the IVs as exposure factors were intersected with the outcome data and used as the final SNPs for performing MR analysis. (Page. 6, Line. 121-123) 

10. Response to comment: Line 118: Citation for IVW validation efficacy?

Response: Thanks for your professional reminder. Based on your valuable suggestions we have added the corresponding references PMID: 23863760 and PMID: 26661904. (Page. 7, Line. 133) 

11. Response to comment: Line 119: A justification for 0.05 as a significance threshold may be helpful given the 6 tests for each outcome.

Response: Thank you for reminding me. Based on your valuable suggestions we have added an explanation for the definition of the p-value, which is added as follows: In this study, six tests were calculated for the MR analysis of each exposure factor, and the IVW test results were mainly used as the main result to determine the causal relationship, and the rest of the results could not infer the causal effect for the time being, so this study in the analysis of the results was considered that p<0.05 for IVW was considered to be statistically significant. (Page. 7, Line. 133-138) 

12. Response to comment: Line 145-147: With such large sample sizes, large F-statistics are not surprising and may not give much insight into instrument strength

Response: Thanks for your reminding. As you said, the validation efficacy of the F-statistic values may be reduced when the sample is studied. However, we have reviewed a lot of literature and have not been able to find an alternative way to calculate it for the time being. Therefore, at the end of the discussion section, we add the limitations of the validation effectiveness of the F-statistics. We modified the following: At the same time, the data of F-statistic value in the study of large sample data analysis will be very large, but the corresponding validation effectiveness will be reduced. (Page. 16, Line. 290-292) 

13. Response to comment: Line 173: positive causal relationships

Response: Your advice is very helpful. Based on your suggestion, we modified it to positive causal relationships. (Page. 11, Line. 206) 

14. Response to comment: Line 176: Non-significant findings do not necessarily suggest a lack of a causal relationship but may be due to lack of statistical power or sufficient instruments to identify a true causal relationship.

Response: The explanation you provided is very helpful to us. As you said, we have added an explanation of the negative causality in the text at the end of the first paragraph of the Discussion section. The modification is as follows: It may be due to lack of statistical power or sufficient IVs to identify a true causal relationship. (Page. 14, Line. 242-243) 

15. Response to comment: Line 202: indicators and TB using two-sample mendelian randomization.

Response: Your advice is very helpful. Based on your suggestion, we modified it to two-sample MR. (Page. 13, Line. 227) 

16. Response to comment: Line 202-205: These MR methods provide linear estimates of effect, not correlations.

Response: Thank you for the reminder. Based on your suggestion, we modified it to linear estimates of effect. (Page. 13, Line. 232) 

We tried our best to improve the manuscript and made some changes in the manuscript. In addition, We have revised and added to the structure of the results and the discussion section of the manuscript as a whole based on the valuable comments of the reviewers. 

We appreciate for Editor and Reviewers’ warm work earnestly, and hope that the correction will meet with approval.

Once again, thank you very much for your comments and suggestions.

---

## [Decision Letter · Decision Letter 1]

6 Sep 2023

PONE-D-23-09863R1Obesity-related indicators and tuberculosis a Mendelian randomization studyPLOS ONE

Dear Dr. Huang,

Thank you for submitting your manuscript to PLOS ONE. After careful consideration, we feel that it has merit but does not fully meet PLOS ONE’s publication criteria as it currently stands. Therefore, we invite you to submit a revised version of the manuscript that addresses the points raised during the review process. Please pay extra attention to the comments and suggestions by reviewer #3.

We look forward to receiving your revised manuscript.

Kind regards,

Chunyu Liu, PhD

Academic Editor

PLOS ONE

Reviewers' comments:

Reviewer's Responses to Questions

**Comments to the Author**

1. If the authors have adequately addressed your comments raised in a previous round of review and you feel that this manuscript is now acceptable for publication, you may indicate that here to bypass the “Comments to the Author” section, enter your conflict of interest statement in the “Confidential to Editor” section, and submit your "Accept" recommendation.

Reviewer #2: (No Response)

Reviewer #3: (No Response)

2. Is the manuscript technically sound, and do the data support the conclusions?

Reviewer #2: Yes

Reviewer #3: No

3. Has the statistical analysis been performed appropriately and rigorously? 

Reviewer #2: Yes

Reviewer #3: No

4. Have the authors made all data underlying the findings in their manuscript fully available?

Reviewer #2: Yes

Reviewer #3: No

5. Is the manuscript presented in an intelligible fashion and written in standard English?

Reviewer #2: Yes

Reviewer #3: No

6. Review Comments to the Author

Reviewer #2: Line 52-53: Remove “As we all know”

Lines 133-138: This wording is a bit misleading. remove “The rest of the results could not infer the causal effect for the time being”. Enough to say that IVW was the main method and therefore P<0.05.

Lines 228-230: “unbiased estimates of the three assumptions” does not make sense. The assumptions are not being estimated. How about. “We used the results of IVW as a criterion for judging

228 the causal relationship between exposure and outcome, while the other methods assessed the validity of the exclusion assumption(32).”

Reviewer #3: The authors used two-sample MR to infer causal association between obesity-related indicators and TB. They observed a causal relationship between waist circumference and TB, and between hip circumference and TB. While these findings are intriguing, further analyses are necessary to validate the robustness and reliability of the results. However, it is essential to point out that the manuscript contains several instances of inappropriate usage of professional terminology and poorly structured sentences, which may impact the clarity and coherence of the content. Therefore, it is advisable to reevaluate the wording and flow of the paper to ensure its accuracy and comprehensibility.

1. The authors tested BMI, waist circumference, hip circumference and waist to hip ratio, it is better to specify a threshold for multiple testing.

2. The authors employed the "PhenoScanner" database to identify and exclude genetic variants associated with potential confounders in their analyses. However, it is crucial to recognize that this approach might not entirely differentiate between horizontal and vertical pleiotropy, as only the former would introduce bias in Mendelian Randomization (MR) studies. Additionally, for many genetic variants, their precise biological functions remain unknown, adding further complexity to the analysis. To enhance the robustness of instrumental variables and strengthen the genetic underpinnings of the study, the authors should consider employing alternative techniques for identifying horizontal pleiotropic single nucleotide polymorphisms (SNPs).

3. The authors accessed tuberculosis (TB) genetic data from a Genome-Wide Association Study (GWAS) database. Unfortunately, the manuscript lacks specific details about the name or source of this GWAS database. In addition, to claim that their analysis is a "two-sample" Mendelian Randomization (MR) analysis, the authors must address whether there is any sample overlapping between the TB genetic data obtained from the GWAS database and the sample used for the exposure data. Without this crucial information, the validity of the "two-sample" MR analysis might be compromised. Sample overlap between datasets used in MR analysis can introduce bias and invalidate the results. Hence, the authors should thoroughly investigate and disclose any shared samples or related individuals between the TB genetic data and exposure data. If there is no overlap, they can confidently proceed with the "two-sample" MR analysis, using distinct datasets for the genetic instrument and exposure, enhancing the robustness and reliability of their study. Otherwise, state in discussion as a limitation of this study.

4. If adopting multiple testing threshold, for example 0.05/4=0.0125, then almost all of the results become null results.

5. The number of IVs used for each exposure variable might be relatively small, which could affect the statistical power of the analysis. Power calculation should be included in the analysis.

6. The analysis conducted in the study did not fully address the main hypothesis of the research. There are several strong risk factors associated with both obesity and tuberculosis, which were not adequately accounted for in the analysis. For instance, lower socioeconomic status is known to increase the risk of tuberculosis and may also contribute to higher rates of obesity due to limited access to healthy foods and opportunities for physical activity. Similarly, smoking is a well-established risk factor for tuberculosis and can also impact body weight, potentially leading to obesity or underweight, depending on smoking habits. Furthermore, diabetes is linked to both obesity and an increased risk of tuberculosis due to its impact on the immune response and susceptibility to infections. To comprehensively examine how these factors may affect tuberculosis, the authors need to incorporate more sophisticated methods beyond simply removing the associated single nucleotide polymorphisms (SNPs) from the analysis. Two-step Mendelian Randomization (MR), multivariable MR, or mediation analysis could be valuable approaches to address the influence of these confounding factors. Two-step MR involves first estimating the effects of the risk factors on the exposure variables (e.g., obesity-related indicators) and then using these estimates to assess the effect of the exposure on the outcome (tuberculosis). Multivariable MR allows for simultaneous assessment of multiple exposures while adjusting for potential confounders. Mediation analysis can help explore whether the effects of the exposure variables on tuberculosis are mediated through other risk factors, such as socioeconomic status, smoking, or diabetes.

By incorporating these methods, the authors can gain a more comprehensive understanding of the relationships between obesity-related indicators, potential confounding factors, and tuberculosis. This will enhance the validity and interpretation of the study's findings and provide more insights into the causal pathways between obesity-related indicators and tuberculosis risk.

Minor:

1. Line 114: Sample mode -> simple mode

2. Line 119 is not clear: “In addition, p <0.05 for each test was considered to be statistically different.”

3. If the a number is very small, use scientific digits to show the numbers. For example, write 0.00023 as 2.3e-05 rather than 0.000.

4. The authors should exercise caution in making strong claims about causality derived from observational data throughout the manuscript. While significant genetic associations may indicate a potential for causality, definitive claims of causation should be avoided. Observational studies, including Mendelian Randomization (MR), can provide valuable insights into causal relationships, but they inherently carry limitations that prevent absolute causality conclusions.

5. The term "horizontal multiplicity" is not a standard term in the context of genetics or statistical analyses. I would suggest to change it throughout the manuscript.

6. Line 188 is not clear: the author stated there were no abnormal outliers.

7. A recent MR study reported a causal relationship between waist circumference and type 2 diabetes, which led to an increased risk for development. This sentence appears to be incomplete

8. The MR-egger test for the hip circumference yielded an extremely large OR and a wide confidence interval, how should that be interpreted? Concluding this result had the same trend in effect was not accurate.

7. PLOS authors have the option to publish the peer review history of their article (what does this mean?). If published, this will include your full peer review and any attached files.

Reviewer #2: **Yes: **Daniel DiCorpo

Reviewer #3: No

---

## [Author Response · Author response to Decision Letter 1]

18 Oct 2023

Dear PhD. Chunyu Liu, 

Thank you for your letter and for the reviewers’ comments concerning our manuscript entitled “Obesity-related indicators and tuberculosis a Mendelian randomization study” (ID: PONE-D-23-09863R1). Those comments are all valuable and very helpful for revising and improving our paper, as well as the important guiding significance to our researches. We have studied comments carefully and have made correction which we hope meet with approval. In tracking the revised manuscript, we yellow the revised and supplemented parts. 

We have fully addressed each concern and hope that this revised manuscript is now acceptable. The main corrections in the paper and the responds to the reviewer’s comments are as following:

Responds to the comments to the Author: 

The above reviewers' comments evaluated the article in terms of content, language, statistical methods, etc. In many aspects, we still have some deficiencies, and we revised the article again, especially thanks to reviewer 3 for his help to us, his suggestions are very valuable and professional, which made us realize the deficiencies of our study.

Responds to the Reviewer’s comments: 

Reviewer #2:

1. Response to comment: 

Line 52-53: Remove “As we all know”

Lines 133-138: This wording is a bit misleading. remove “The rest of the results could not infer the causal effect for the time being”. Enough to say that IVW was the main method and therefore P<0.05.

Lines 228-230: “unbiased estimates of the three assumptions” does not make sense. The assumptions are not being estimated. How about. “We used the results of IVW as a criterion for judging

228 the causal relationship between exposure and outcome, while the other methods assessed the validity of the exclusion assumption(32).”

Response: Thank you very much for your careful comments, based on your valuable suggestions, we have made changes, thank you again for your hard work.

Reviewer #3: 

The authors used two-sample MR to infer causal association between obesity-related indicators and TB. They observed a causal relationship between waist circumference and TB, and between hip circumference and TB. While these findings are intriguing, further analyses are necessary to validate the robustness and reliability of the results. However, it is essential to point out that the manuscript contains several instances of inappropriate usage of professional terminology and poorly structured sentences, which may impact the clarity and coherence of the content. Therefore, it is advisable to reevaluate the wording and flow of the paper to ensure its accuracy and comprehensibility.

Response to comment: Thank you very much for your comments. We think you are very well versed in the study of MR and your suggestions have given us a better understanding of MR. Your reminder makes us realize that removing confounding factors via PhenoScanner is not accurate enough. In order to minimize the pleiotropy from smoking, type 2 diabetes, and educational attainment, we conducted a multivariable MR analysis with adjustments for genetically predicted of these variables. We have reanalyzed the overall data and have come up with some new conclusions that give more insight into our study. Thanks again for your dedication and guidance.

1. Response to comment: The authors tested BMI, waist circumference, hip circumference and waist to hip ratio, it is better to specify a threshold for multiple testing.

Response: Your advice is very important. Based on your valuable suggestions we have increased the threshold for multiple testing. We modified it as follows: “Given that our study evaluates four null hypotheses, we have employed a Bonferroni-corrected Type I error rate of αBonf = 0.0125(0.05/4=0.0125) to address the issue of multiple testing” (Page. 7, Line. 134-136)

2. Response to comment: The authors employed the "PhenoScanner" database to identify and exclude genetic variants associated with potential confounders in their analyses. However, it is crucial to recognize that this approach might not entirely differentiate between horizontal and vertical pleiotropy, as only the former would introduce bias in Mendelian Randomization (MR) studies. Additionally, for many genetic variants, their precise biological functions remain unknown, adding further complexity to the analysis. To enhance the robustness of instrumental variables and strengthen the genetic underpinnings of the study, the authors should consider employing alternative techniques for identifying horizontal pleiotropic single nucleotide polymorphisms (SNPs).

Response: Thank you very much for your professional advice. Your reminder makes us realize that removing confounding factors via PhenoScanner is not accurate enough. To make our study more reliable, we supplemented multivariate MR .We modified it as follows: “In order to minimize the pleiotropy from smoking, type 2 diabetes, and educational attainment, we conducted a multivariable MR analysis with adjustments for genetically predicted of these variables.” (Page. 7, Line. 144-146)

3. Response to comment: The authors accessed tuberculosis (TB) genetic data from a Genome-Wide Association Study (GWAS) database. Unfortunately, the manuscript lacks specific details about the name or source of this GWAS database. In addition, to claim that their analysis is a "two-sample" Mendelian Randomization (MR) analysis, the authors must address whether there is any sample overlapping between the TB genetic data obtained from the GWAS database and the sample used for the exposure data. Without this crucial information, the validity of the "two-sample" MR analysis might be compromised. Sample overlap between datasets used in MR analysis can introduce bias and invalidate the results. Hence, the authors should thoroughly investigate and disclose any shared samples or related individuals between the TB genetic data and exposure data. If there is no overlap, they can confidently proceed with the "two-sample" MR analysis, using distinct datasets for the genetic instrument and exposure, enhancing the robustness and reliability of their study. Otherwise, state in discussion as a limitation of this study.

Response: Your advice is very valuable. We are very sorry that due to our negligence, we did not provide a detailed introduction of exposure and outcome data. We have added a description of the exposure and outcome databases in the Materials and Methods section and applied the corresponding references. It was clarified that there is no sample overlap between exposure and outcome. We modified it as follows: “In this study, genetic variation data for obesity-related indicators, including BMI, waist circumference, hip circumference, waist-to-hip ratio, were obtained from the Genetic Investigation of Anthropometric Traits (GIANT) Consortium. The genetic variation data associated with TB was obtained from the Finngen biobank. All the data were sourced from distinct cohorts of European ancestry, with exposures and outcomes stemming from different study, devoid of overlap.” (Page. 5, Line. 86-95)

4. Response to comment: If adopting multiple testing threshold, for example 0.05/4=0.0125, then almost all of the results become null results.

Response: Thank you very much for your advice, which is very professional. According to your reminder, when we initially screened the instrumental variables, smoking, diabetes and education level were regarded as the main confounding factors affecting tuberculosis. Therefore, we re-conducted the two-sample MR analysis and finally determined that waist circumference was in line with the p-values corrected by multiple tests.We modified it as follows: “Based on our comprehensive literature search, we have identified three TB-related factors as potential confounding variables that could impact the genuine causal relationship between obesity-related indicators and TB: smoking, type 2 diabetes, and educational attainment.” (Page. 5, Line. 90-94, Table 3)

5. Response to comment: The number of IVs used for each exposure variable might be relatively small, which could affect the statistical power of the analysis. Power calculation should be included in the analysis.

Response: According to your valuable suggestions, we have supplemented the power. We modified it as follows: “The power corresponding to waist circumference in this study was calculated using an online tool (https://shiny.cnsgenomics.com/mRnd/). The relevant parameters are set as follows: type-I error rate is 0.05, R2 is 0.0002, based on the overall sample size, the final power we get is 84%. This result suggest that this study has sufficient power to explore the causal relationship between waist circumference and TB.” (Page. 13, Line. 226-231)

6. Response to comment: The analysis conducted in the study did not fully address the main hypothesis of the research. There are several strong risk factors associated with both obesity and tuberculosis, which were not adequately accounted for in the analysis. For instance, lower socioeconomic status is known to increase the risk of tuberculosis and may also contribute to higher rates of obesity due to limited access to healthy foods and opportunities for physical activity. Similarly, smoking is a well-established risk factor for tuberculosis and can also impact body weight, potentially leading to obesity or underweight, depending on smoking habits. Furthermore, diabetes is linked to both obesity and an increased risk of tuberculosis due to its impact on the immune response and susceptibility to infections. To comprehensively examine how these factors may affect tuberculosis, the authors need to incorporate more sophisticated methods beyond simply removing the associated single nucleotide polymorphisms (SNPs) from the analysis. Two-step Mendelian Randomization (MR), multivariable MR, or mediation analysis could be valuable approaches to address the influence of these confounding factors. Two-step MR involves first estimating the effects of the risk factors on the exposure variables (e.g., obesity-related indicators) and then using these estimates to assess the effect of the exposure on the outcome (tuberculosis). Multivariable MR allows for simultaneous assessment of multiple exposures while adjusting for potential confounders. Mediation analysis can help explore whether the effects of the exposure variables on tuberculosis are mediated through other risk factors, such as socioeconomic status, smoking, or diabetes.

By incorporating these methods, the authors can gain a more comprehensive understanding of the relationships between obesity-related indicators, potential confounding factors, and tuberculosis. This will enhance the validity and interpretation of the study's findings and provide more insights into the causal pathways between obesity-related indicators and tuberculosis risk. 

Response: Your suggestion is very valuable to our research. Thank you very much for your guidance. According to your valuable suggestions, we supplemented the multivariate MR Analysis. We were unable to find a suitable dataset for socioeconomic status in the GWAS database, so we substituted educational attainment. We analyzed the effects of confounding factors such as smoking, diabetes, and education level on our findings using multivariate MR. We modified it as follows: “Simultaneously, in the multivariable MR analysis adjusting for genetically predicted potential confounding factors, the results remained steadfast (S5-8 Tables).” (Page. 9-10, Line. 188-190)

Minor Remarks:

1. Response to comment: Line 114: Sample mode -> simple mode

Response: Thank you very much for your reminding, we have corrected it. (Page. 6, Line. 127)

2. Response to comment: Line 119 is not clear: “In addition, p <0.05 for each test was considered to be statistically different.” 

Response: Thank you for your valuable advice. We modified it as follows: “Given that our study evaluates four null hypotheses, we have employed a Bonferroni-corrected Type I error rate of αBonf = 0.0125(0.05/4=0.0125) to address the issue of multiple testing. (Page. 7, Line. 134-136)

3. Response to comment: If the a number is very small, use scientific digits to show the numbers. For example, write 0.00023 as 2.3e-05 rather than 0.000.

Response: Thank you very much for reminding us that we have made some adjustments in the writing of numbers in the article. 

4. Response to comment: The authors should exercise caution in making strong claims about causality derived from observational data throughout the manuscript. While significant genetic associations may indicate a potential for causality, definitive claims of causation should be avoided. Observational studies, including Mendelian Randomization (MR), can provide valuable insights into causal relationships, but they inherently carry limitations that prevent absolute causality conclusions.

Response: Thank you for reminding me. Based on your valuable comments, we have revised the article as a whole. According to the analysis results, we found that waist circumference is likely to be potential risk factors for tuberculosis. We modified it as follows: “In summary, we determined a causal relationship between waist circumference and TB using two-sample MR analysis, which suggested that waist circumference is likely to be potential risk factors.” (Page. 17, Line. 304-306)

5. Response to comments: The term "horizontal multiplicity" is not a standard term in the context of genetics or statistical analyses. I would suggest to change it throughout the manuscript.

Response: Thank you very much for pointing out our problem. We changed it throughout the article and replaced it with pleiotropy. (Page. 9, Line. 182-187)

6. Response to comments: Line 188 is not clear: the author stated there were no abnormal outliers.

Response: We are very sorry that we did not express ourselves clearly. We would like to show that there are no extreme outliers for the SNPs selected in this study through the MR-PRESSO results. We'll change this to “Furthermore, the p-values for MR-Egger intercept and MR-PRESSO global test were > 0.05 for the pleiotropy test, indicating that the SNPs included had no pleiotropy and no abnormal outliers were found for the SNPS selected in this study(Table 3).” (Page. 9, Line. 185-188)

7. Response to comments: A recent MR study reported a causal relationship between waist circumference and type 2 diabetes, which led to an increased risk for development. This sentence appears to be incomplete.

Response: We are very sorry that we did not express ourselves clearly due to our negligence. We'll change this to “A recent MR study reported a causal relationship between waist circumference and type 2 diabetes, and type 2 diabetes also leads to an increased risk of TB”. (Page. 14, Line. 255-257) 

8. Response to comments: The MR-egger test for the hip circumference yielded an extremely large OR and a wide confidence interval, how should that be interpreted? Concluding this result had the same trend in effect was not accurate.

Response: Your advice is very valuable. As for the problem of too large confidence interval, we included it in the limitations of this study. We believe that it may be related to the selected instrumental variables, and subsequent increases in their number may solve this problem. In addition, following your reminder, we realized that it would be imprecise to conclude that this method has the same trend as other methods, so we removed this conclusion from the manuscript. We'll change this to “In additon, some excessive confidence intervals were obtained in the MR analysis, and the sample size and IVs need to be expanded in the future.”. (Page. 16, Line. 298-300) 

We tried our best to improve the manuscript and made some changes in the manuscript. In addition, We have revised and added to the structure of the results and the discussion section of the manuscript as a whole based on the valuable comments of the reviewers. 

We appreciate for Editor and Reviewers’ warm work earnestly, and hope that the correction will meet with approval.

Once again, thank you very much for your comments and suggestions.

---

## [Decision Letter · Decision Letter 2]

16 Jan 2024

Obesity-related indicators and tuberculosis a Mendelian randomization study

PONE-D-23-09863R2

Dear Dr. Huang,

We’re pleased to inform you that your manuscript has been judged scientifically suitable for publication and will be formally accepted for publication once it meets all outstanding technical requirements.

Kind regards,

Chunyu Liu, PhD

Academic Editor

PLOS ONE

Additional Editor Comments (optional):

Reviewers' comments:

Reviewer's Responses to Questions

**Comments to the Author**

1. If the authors have adequately addressed your comments raised in a previous round of review and you feel that this manuscript is now acceptable for publication, you may indicate that here to bypass the “Comments to the Author” section, enter your conflict of interest statement in the “Confidential to Editor” section, and submit your "Accept" recommendation.

Reviewer #1: All comments have been addressed

Reviewer #4: (No Response)

2. Is the manuscript technically sound, and do the data support the conclusions?

Reviewer #1: Yes

Reviewer #4: Yes

3. Has the statistical analysis been performed appropriately and rigorously? 

Reviewer #1: Yes

Reviewer #4: Yes

4. Have the authors made all data underlying the findings in their manuscript fully available?

Reviewer #1: Yes

Reviewer #4: Yes

5. Is the manuscript presented in an intelligible fashion and written in standard English?

Reviewer #1: Yes

Reviewer #4: Yes

6. Review Comments to the Author

Reviewer #1: (No Response)

Reviewer #4: The authors have adequately addressed the previous comments from reviewers in the revised manuscript. I believe the paper merits publication in its current form. Below, I have two minor comments that could potentially enhance the manuscript, particularly concerning the newly added multivariate MR analysis.

1. While the multivariate MR approach effectively addresses comment 6 from reviewer 3, it is acknowledged that adjusting for socioeconomic status presents a challenge due to its absence. However, utilizing education as a substitute for socioeconomic status may compromise the analysis. The author may consider disclosing this limitation in the discussion.

2. The multivariate MR to demonstrate the potential causal association between waist circumference and TB was not significantly mediated by other potential factors, including education, smoking, and diabetes. To some extent, it is surprising that none of the known factors mediated even a portion of this association, which is fine. The authors may additionally consider these two potential options to improve upon this point: they could either add some discussion about the lack of a mechanism for the observed association or conduct a brief multivariate MR analysis to explore whether this association is mediated by immune or other relevant biological pathways.

7. PLOS authors have the option to publish the peer review history of their article (what does this mean?). If published, this will include your full peer review and any attached files.

Reviewer #1: No

Reviewer #4: No

---

## [Editor Report · Acceptance letter]

22 Mar 2024

PONE-D-23-09863R2 

PLOS ONE

Dear Dr. Huang, 

I'm pleased to inform you that your manuscript has been deemed suitable for publication in PLOS ONE. Congratulations! Your manuscript is now being handed over to our production team.

Kind regards, 

on behalf of

Dr. Chunyu Liu 

Academic Editor

PLOS ONE